# Scalable Algorithms for Maximizing Spatiotemporal Range Sum and Range Sum Change in Spatiotemporal Datasets

**Woosung Choi [1], Soon-Young Jung [1], Jaehwa Chung [2], Kyeong-Seok Hyun [1] and Kinam Park [3],***

[1]  Department of Computer Science, Korea University, Seoul 02841, Korea; ws_choi@korea.ac.kr (W.C.); jsy@korea.ac.kr (S.-Y.J.); ks_hyun@korea.ac.kr (K.-S.H.)
[2]  Department of Computer Science, Korea National Open University, Seoul 03087, Korea; jaehwachung@knou.ac.kr
[3]  Creative Information and Computer Institute, Korea University, Seoul 02841, Korea
*  Correspondence: spknn@korea.ac.kr; Tel.: +82-10-9194-8289

**Abstract:** In this paper, we introduce the three-dimensional Maximum Range-Sum (3D MaxRS) problem and the Maximum Spatiotemporal Range-Sum Change (MaxStRSC) problem. The 3D MaxRS problem tries to find the 3D range where the sum of weights across all objects inside is maximized, and the MaxStRSC problem tries to find the spatiotemporal range where the sum of weights across all objects inside is maximally increased. The goal of this paper is to provide efficient methods for data analysts to find interesting spatiotemporal regions in a large historical spatiotemporal dataset by addressing two problems. We provide a mathematical explanation for each problem and propose several algorithms for them. Existing methods tried to find the optimal region over two-dimensional datasets or to monitor a burst region over two-dimensional data streams. The majority of them cannot directly solve our problems. Although some existing methods can be used or modified to solve the 3D MaxRS problems, they have limited scalability. In addition, none of them can be used to solve the MaxStRS-RC problem (a type of MaxStRSC problem). Finally, we study the performance of the proposed algorithms experimentally. The experimental results show that the proposed algorithms are scalable and much more efficient than existing methods.

**Keywords:** MaxStRSC; 3D MaxRS; plane-sweep; scalability

## 1. Introduction

Technological advances in mobile devices, location tracking, and wireless communication lead to the emergence of new types of services, such as location-based social networking services. Nowadays, a vast amount of user-generated spatiotemporal data has been collected from these services. Analyzing these spatiotemporal data often provides insights into understanding customers' behaviors in the real world. For example, suppose that data analysts work for a coffeehouse chain that has over 2000 retail stores across the globe. In addition, suppose that they obtain a large historical dataset by having collected geo-tagged posts that mentioned their coffeehouse from several Location-Based Social Network Services (LBSNSs) to analyze customer satisfaction. Each collected object $o$ is represented by a tuple of the form $< x, y, t, w >$, where $(x, y, t)$ is the spatiotemporal point at which $o$ is posted, and $w$ is the weight of $o$.

In this scenario, the following queries may help develop a marketing strategy.

- *Query 1*. "find the (1 km $\times$ 1 km $\times$ 1 h) spatiotemporal range which maximizes the number of objects inside."

- *Query 2.* "find the (1 km × 1 km × 1 h) spatiotemporal range where the number of objects inside is maximally increased within one hour.

Figure 1 describes query examples on an example dataset. We draw only four representative ranges (i.e., *A*, *B*, *C*, and *D*), but note that there are an infinite number of (1 km × 1 km × 1 h) ranges in the entire space. The result of query 1 is *B* because no other spatiotemporal ranges contain more than five objects. However, the result of query 2 is not *B* but *C*. Let us draw the 'previous' range below each range, as shown in Figure 1. For example, we draw $A_{prev}$ below *A* so that $A_{prev}$ is adjacent to *A*. Observe that the number of objects inside *A* is increased by one compared to that inside $A_{prev}$. As shown in Figure 1b, *C* is the region where the number of posts inside is maximally increased.

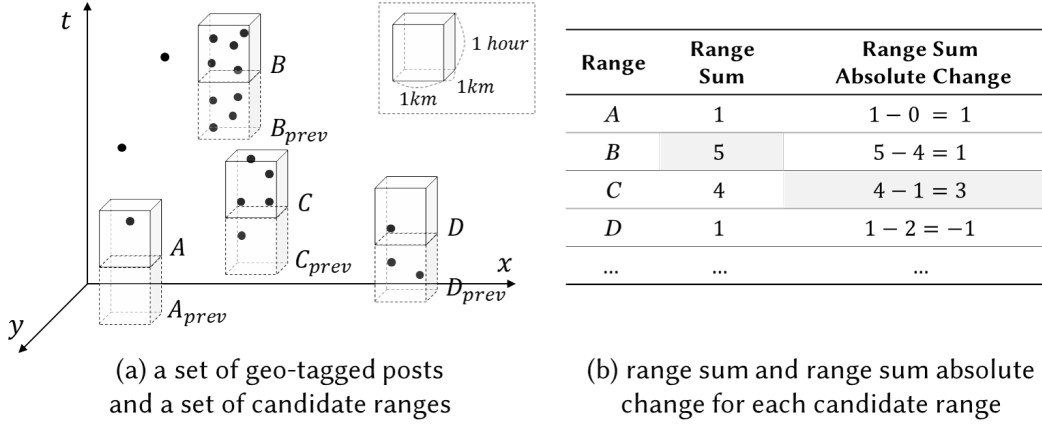

| Range | Range Sum | Range Sum Absolute Change |
|:-----:|:---------:|:-------------------------:|
| *A* | 1 | $1 - 0 = 1$ |
| *B* | 5 | $5 - 4 = 1$ |
| *C* | 4 | $4 - 1 = 3$ |
| *D* | 1 | $1 - 2 = -1$ |
| ... | ... | ... |

(a) a set of geo-tagged posts and a set of candidate ranges

(b) range sum and range sum absolute change for each candidate range

**Figure 1.** An example dataset and query examples.

It is helpful for data analysts to use these queries to understand customers' behavior. In addition, they can provide more valuable insights into customers' satisfaction with a slight modification. For example, suppose that we set the weight of each geo-tagged post by quantifying the emotional state of the poster (e.g., +1 for positive and −1 for negative) using existing sentiment analysis methods. Then, data analysts can carry out case-studies for marketing research by using the following queries:

- *Query 3.* "find the (1 km × 1 km × 1 h) spatiotemporal range which maximizes the number of all positive geo-tagged posts inside."
- *Query 4.* "find the (1 km × 1 km × 1 h) spatiotemporal range where the number of negative geo-tagged posts inside is maximally increased within one hour."

If we plan to host an event for marketing in a restricted area and time, these queries may be useful to prepare this event. The goal of this paper is to provide efficient methods for data analysts to find such interesting spatiotemporal regions in a large historical spatiotemporal dataset. We formulate two mathematical problems: the maximum three-dimensional range-sum (3D MaxRS) problem and the maximum spatiotemporal range-sum change (MaxStRSC) problem. The 3D MaxRS problem tries to find the spatiotemporal range where the sum of weights across all objects inside is maximized. On the other hand, the MaxStRSC problem tries to find the spatiotemporal range where the sum of weights across all objects inside is maximally increased.

Recently, many researchers have proposed methods for analyzing spatiotemporal data to support decision-making processes. Existing methods try to find the optimal region over 2D datasets or to monitor a burst region over 2D data streams. However, the majority of them cannot directly solve our problems. Although some existing methods can be used or modified to solve the 3D MaxRS problems, they have limited scalability. In addition, none of them can be used to solve the MaxStRS-RC problem (a type of MaxStRSC problem). To our knowledge, this is the first research addressing the 3D MaxRS problem and the MaxStRSC problem in historical spatiotemporal datasets.

We summarize our contribution as follows: (1) We introduce two problems (3D MaxRS, MaxStRSC), which try to find interesting spatiotemporal regions in a large historical spatiotemporal dataset. To our knowledge, this is the first research that studies these problems over large-scale historical spatiotemporal datasets. (2) We provide a mathematical explanation and propose several scalable algorithms for each problem. (3) We experimentally study the performance of the proposed algorithms. We extend existing methods so that they can handle the 3D MaxRS problem or the MaxStRSC problem for comparison. The experimental results show that our algorithms are much more efficient and scalable than existing methods.

We organize the rest of this paper as follows: We give a formal definition for each problem in Section 2. We survey the related works in detail in Section 3. Then, we describe our algorithms for the 3D MaxRS problem (Section 4) and the MaxStRSC problem (Sections 5 and 6). In Section 7, we discuss the experimental results. In Section 8, we summarize the conclusion of this paper and our future work.

## 2. Preliminaries

In this section, we give a formal definition of the 3D MaxRS problem and the MaxStRSC problem.

### 2.1. 3D MaxRS Problem

Let us consider a set $O$ of spatiotemporal objects whose cardinality is too large to fit into main memory. An object $o \in O$ is defined as a tuple $< x, y, t, w >$, where $(x, y, t)$ represents the spatiotemporal point, and $w$ is the weight of $o$.

We now introduce some useful notations to express geometric objects compactly. We use $c(p)$ to denote the $a \times b \times \tau$ axis-parallel cuboid centered at a point $p = (x, y, t) \in P$, where $P$ is the entire spatiotemporal space. In addition, for a cuboid $c$, we denote the set of objects inside $c$ by $O(c)$ (i.e., $O(c) = \{o | o \in O$ and $(o.x, o.y, o.t)$ is inside $c\}$). Without loss of generality, an object on the boundary of a cuboid is not said to be 'inside' the cuboid.

Using these notations, we define the three-dimensional Maximum Spatiotemporal Range Sum (3D MaxRS) problem as follows:

**Definition 1.** *(3D MaxRS problem). Given O, the size $a \times b \times \tau$ of a query cuboid, find the set of all points $p^* \in P$ such that:*

$$p^* = \arg\max_{p \in P} \left( \sum_{o \in O(c(p))} o.w \right),$$

*where P is the entire spatiotemporal space.*

As its name implies, the 3D MaxRS problem is a three-dimensional version of the 2D MaxRS problem [1,2], a well-known problem studied in computational geometry community.

### 2.2. MaxStRSC Problem

We introduce another notation before we formulate the maximum range sum change problem. We use $c_{prev}(p)$ to denote the $a \times b \times \tau$ axis-parallel cuboid centered at the point $p_{prev} = (p.x, p.y, p.t - \tau)$, where $p = (x, y, t) \in P$. The Maximum Spatiotemporal Range Sum Change (MaxStRSC) problem is defined as follows:

**Definition 2.** *(MaxStRSC problem). Given O, the size $a \times b \times \tau$ of a query cuboid, and a function change: $(\mathbb{R} \times \mathbb{R}) \to \mathbb{R}$, find the set of all points $p^* \in P$ such that:*

$$p^* = \arg\max_{p \in P} \left( change(\sum_{o \in O(c_{prev}(p))} o.w, \sum_{o \in O(c(p))} o.w) \right)$$

*where P is the entire spatiotemporal space.*

The MaxStRSC tries to find the location $p$ such that the change between the aggregate weight of objects in $O(c_{prev}(p))$ and the aggregate weight of objects in $O(c(p))$ is maximized. The answer to an instance of the MaxStRSC problem depends on the choice of a function *change*. We propose two types of the MaxStRSC problem called Maximum Spatiotemporal Range-Sum Absolute-Change (MaxStRS-AC), and Maximum Spatiotemporal Range-Sum Relative-Change (MaxStRS-RC) as follows:

- *MaxStRS-AC*: The MaxStRSC problem with the absolute change measurement function $change_{abs}$, where $change_{abs}(x_1, x_2) = x_2 - x_1$
- *MaxStRS-RC*: The MaxStRSC problem with the relative change measurement function $change_{rel}$ where $change_{rel}(x_1, x_2) = \frac{x_2 - x_1}{x_1}$. Without loss of generality, we assume that $change_{rel}(x_1, x_2) = 0$ if $x_1 = 0$.

## 3. Related Work

We categorize related works into four groups as follows: (1) Range Aggregate Query, (2) MaxRS and its variants in 2D Space, (3) Max-enclosure Problem in 3D Space, and (4) Burst Detection.

### 3.1. Range Aggregate Query

Suppose that we have a set $O$ of spatiotemporal objects, where each $o \in O$ has weight $o.w$. The range aggregate query [3–8] returns the aggregate value over objects inside the query range. Although it is possible to solve the 3D MaxRS problem by issuing range-aggregate queries, this method needs too high computational costs even for solving the 2D MaxRS problem as described in [1].

### 3.2. MaxRS and Its Variants in 2D Space

For the given set $O$ of weighted objects and the size of a query rectangle (or cuboid), the 2D MaxRS problem tries to find the location of the area where the sum of weights of all objects is maximized. The majority of studies in this category [1,2,9–14] aim to process the 2D MaxRS query on static spatial objects. Choi et al. [1,2] proposed scalable algorithms, by modifying the in-memory plane-sweep algorithm [15], and Zhou et al. [9] proposed an index-based method that solves the MaxRS query. Several variants of the MaxRS problem [10–14] also have been proposed. Some researchers proposed methods [16–20] for continuously monitoring the MaxRS problem over spatial data streams.

The existing methods mentioned above are not targeted at the historical spatiotemporal data analysis. While the input of the 3D MaxRS problem and the MaxStRSC problem is a set of spatiotemporal objects, the input of the MaxRS problem of each study above is either a set of spatial objects or a spatial data stream in a two-dimensional space. Most of them cannot be directly extended to solve the 3D MaxRS problem or the MaxStRSC problem. Although we can use one of some existing algorithms [1,2] as a sub-routine in our method at least for solving the 3D MaxRS problem, it degrades the performance due to redundant computations. We describe this in detail in Section 4.2.3, and Section 7.2.1.

### 3.3. Max-Enclosure Problem in 3D Space

We found one study [15] that can be directly applied to solve the 3D MaxRS problem. Subhas C. Nandy and Bhaswar B. Bhattacharya [15] proposed an in-memory algorithm for the max-enclosing cuboid problem, a special case of the 3D MaxRS problem where every object has weight 1. The main idea of their approach is to take the projections of all the input objects on the $xy$-plane and compute the set $S$ of candidate $(x, y)$ coordinates. Although their method is simple to implement, it has the following drawbacks: (1) it needs an expensive preprocessing to compute all the potential $(x, y)$ coordinates; (2) it has limited scalability since large amounts of points can be created in the first phase as the data size increases. We extend it so that it can handle the 3D MaxRS problem and externalize

it by employing STR-tree [21] for better scalability, but the experimental results show that it is much more inefficient than the proposed methods.

*3.4. Burst Detection*

Some existing work [22,23] tried to identify burst regions (hot spots), which is similar to our MaxStRS-AC problem. For example, Mathioudakis et al. [22] propose the spatial burst problem, which seeks to find 'spatial bursts' where related documents exhibit a surge. They tried to find 'burst cells', where notable events happen over a grid. Conceptually, it may seem similar to our works, but they are different. The answer to the MaxStRSC problem can be an arbitrary region that does not need to be one of the pre-defined cells, unlike that of the spatial burst problem of [22].

Recently, Feng et al. [23] proposed the continuous bursty region detection (SURGE) problem, which tries to monitor a bursty region from a stream of spatial objects. For the given size $a \times b$ of a query rectangle, a bursty region is the rectangular region of size $a \times b$ with the maximum burst score. The burst score of a rectangular region $r$ is defined over two consecutive temporal windows $W_{prev}, W_{current}$. The higher the change of the aggregate value of objects inside $r$ for $W_{prev}$ between that of objects inside $r$ for $W_{current}$, the higher tends to be the burst score of $r$ over $W_{prev}, W_{current}$. Unlike studies for the MaxRS problem or its variations, Feng et al. [23] focuses on the change of the aggregate value of the region between two consecutive temporal windows.

Although The SURGE problem is similar conceptually to the MaxStRS-AC problem, they are different. They assume a stream environment, so it is not feasible to use their solution to analyze the large historical spatiotemporal dataset. We can extend it to solve the MaxStRS-AC problem, but it has limited scalability (see Section 7.3). In addition, it is impossible to extend the algorithm for the SURGE problem to solve the MaxStRS-RC problem (see Section 7.1).

## 4. Algorithms for 3D MaxRS

We solve the 3D MaxRS problem by transforming it into an equivalent problem with a finite set of objects to be examined. We present an alternative definition of the 3D MaxRS problem and show how to transform the problem in Section 4.1. Then, we describe our algorithms for the 3D MaxRS in Sections 4.2 and 4.3.

*4.1. Alternative Definition of 3D MaxRS*

In this subsection, we show how to transform the 3D MaxRS problem into an equivalent problem by generalizing the idea of [15] to 3D space. We first introduce notations used to define the new version of the 3D MaxRS problem:

**Definition 3.** *(Weighted Cuboid) For an object $o \in O$, the weighted cuboid $c(o)$ is defined as the axis-parallel cuboid which is centered at $(o.x, o.y, o.t)$, whose size is $a \times b \times \tau$, and whose weight is the same as $o.w$. We denote the weight of a cuboid $c(o)$ by $c.w$. In addition, for the set $O$ of objects, the set $C$ of weighted cuboids is defined as $\{c(o) | o \in O\}$.*

**Definition 4.** *(Point-Weight). Let $C$ be the set of weighted cuboids. For a point $p$ in $P$, the point-weight of $p$ is defined as $\sum_{c \in C(p)} c.w$, where $C(p) = \{c | c \in C \text{ and } c \text{ contains } p\}$, and is denoted by $p.w$.*

**Definition 5.** *(Weighted Cuboid Partition) For the set $C$ of weighted cuboids, a weighted cuboid partition $CP$ is a set of disjoint, weighted cuboids satisfying the following conditions:*

1. *$CP$ is a set of disjoint weighted cuboids, that is, any pair of cuboids in $CP$ are not overlapping.*
2. *The geometric union of cuboids in $CP$ is the same as the geometric union of cuboids in $C$.*
3. *For each cuboid $c \in CP$, the point-weight of any $p \in c$ is the same as $c.w$.*

Figure 2 gives an example of these new concepts. Figure 2a shows the set of objects $O = \{o_1, o_2\}$, where each object has weight 1. As illustrated in Figure 2b, we create a corresponding weighted cuboid for each object. Let us denote the set of created cuboids by $C = \{c(o_1), c(o_2)\}$. Figure 2c describes a set $CP$ of seven weighted cuboids, where the weight of the shaded cuboid (i.e., $c^*$) is set to 2, and weights of the other cuboids are all set to 1. Note that $c^*$ is the common intersection area of $c(o_1)$ and $c(o_2)$. We show that $CP$ satisfies three conditions of Definition 5. First, it is trivial that $CP$ satisfies the first and second conditions, as shown in Figure 2c. In addition, $CP$ satisfies the third condition as well: (1) the point-weight of each point $p^*$ in $c^*$ is 2 since $p^*$ is enclosed by $c(o_1)$ and $c(o_2)$; (2) for each cuboid $c$ in $CP \setminus \{c^*\}$, the point-weight of every point $p$ in $c$ is 1 since $p$ is covered by one cuboid. In other words, $CP$ is the valid weighted cuboid partition of the given $C$.

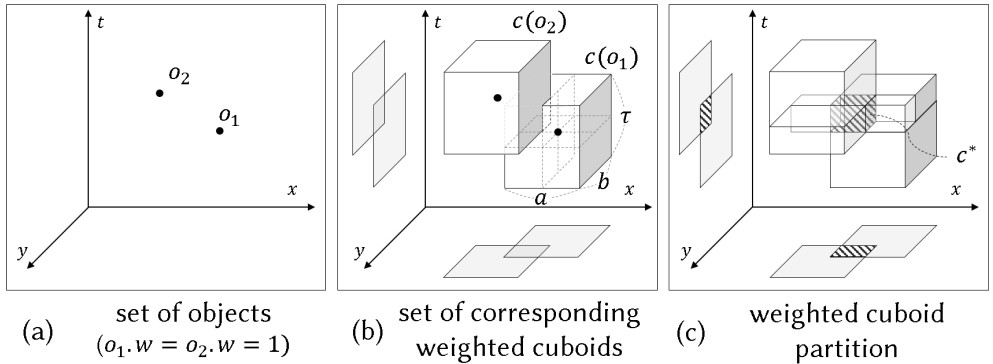

**Figure 2.** An example of the set of weighted cuboids and a weighted cuboid partition.

We now introduce an alternative definition of the 3D MaxRS problem, which is equivalent to the original version as follows:

**Definition 6.** *(Alternative version of the 3D MaxRS problem). For a set C of weighted cuboids, find the set of all cuboids $c^* \in CP$ such that $c^* = \arg\max_{c \in CP} (c.w)$, where CP is a weighted cuboid partition of C.*

If we can obtain a weighted cuboid partition of $C$, it is easy to solve the alternative version since it enables us to solve the problem with a finite set of cuboids to be examined. Going back to the example of Figure 2, suppose that we have an instance of the original 3D MaxRS problem with input $O = \{o_1, o_2\}$, and $a \times b \times \tau$ of a query cuboid. Then, the corresponding input for the alternative version is $C = \{c(o_1), c(o_2)\}$. It is easy to obtain the answer to the alternative version by selecting the set of cuboids with the maximum weight (i.e., $\{c^*\}$) among all cuboids in the provided weighted cuboid partition $CP$.

We now show the equivalence of the two versions with the example of Figure 2. For any point $p \in P$, $p$ is one of: (1) not covered by any cuboid in $CP$, (2) inside a cuboid in $CP \setminus \{c^*\}$, and (3) inside $c^*$. Observe that the point-weight is smaller than $c^*.w$ in the first case ($p.w = 0$) and second case ($p.w = 1$). Thus, every point $p^*$ inside $c^*$ belongs to the answer set of the alternative version because it has the maximum point-weight ($p^*.w = 2$).

### 4.2. Nested Plane-Sweep Algorithm

If we obtain a weighted cuboid partition of $C$, it is easy to obtain the set of optimal cuboids. In this section, we present a nested plane-sweep (NPS) algorithm which solves the transformed 3D MaxRS problem by computing a weighted cuboid partition with the plane-sweep paradigm. Throughout this section, we use the problem instance shown in Figure 2 as a running example to explain the algorithm. We show how the NPS algorithm eventually finds the shaded cuboid in Figure 2c.

### 4.2.1. The Outline of the NPS Algorithm

As its name implies, the NPS algorithm uses the plane-sweep paradigm to solve the problem. To illustrate the idea, let us consider an imaginary plane perpendicular to the t-axis, as shown in Figure 3. While sweeping the plane from bottom to top across the entire space, the algorithm performs some geometric operations whenever the plane meets the top or bottom rectangle of a cuboid. We call such rectangles event rectangles and the imaginary plane as sweep-plane. In addition, we say a cuboid $c$ is active if and only if the sweep-plane has encountered the bottom of $c$ but has not entirely passed over the top of $c$ yet. From now on, we assume that each object in $O$ has different t-axis value for the sake of simplicity. The algorithm terminates after a sweep-plane encounters the $(2 * |C|)$-th (or $(2 * |O|)$-th) event rectangle.

Let us explain terms related to this approach with the example of Figure 2b. Since we sweep the plane from bottom to top, the first event rectangle encountered by the sweep-plane is $r_1$, the bottom of $c_1$ (at $t = t_1$) as shown in Figure 3a. At this moment, there is only one active cuboid, $c_1$. As shown in Figure 3b, the second event rectangle encountered by the sweep-plane is $r_2$, the bottom of $c_2$ (at $t = t_2$). Then, the set of active cuboids at $t = t_2$ becomes $\{c_1, c_2\}$. When the sweep-plane encounters $r_3$, the top of $c_1$ at $t = t_3$; then, $c_1$ becomes inactive. Thus, the set of active cuboids at $t = t_3$ becomes $\{c_2\}$.

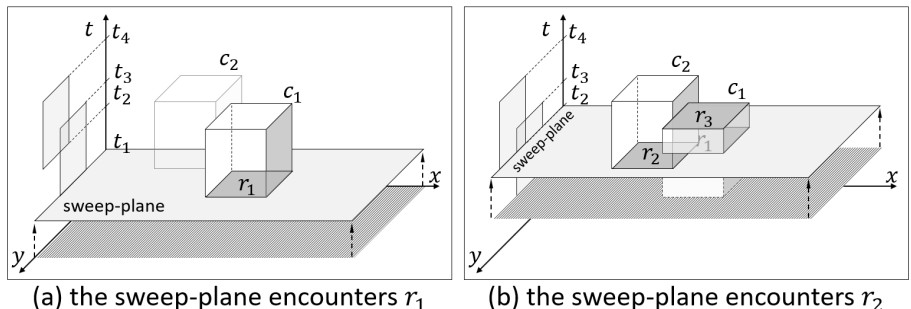

(a) the sweep-plane encounters $r_1$    (b) the sweep-plane encounters $r_2$

**Figure 3.** The plane sweep approach.

The plane-sweep approach enables us to convert the 3D geometric problem into a series of 2D problems. To demonstrate this idea clearly, let us use $r_i$ to denote the i-th event rectangle encountered by the sweep-plane and $t_i$ to denote the t-axis value of the sweep-plane at which the plane encounters $r_i$. Then, the following observation allows us to find the local solution in the subspace $t_i \leq t < t_{i+1}$ by just investigating the 2D plane $t = t_i$.

**Observation 1.** *For any point $p$ in $t_i \leq t < t_{i+1}$, the point-weight of $p$ is the same as the point-weight of $p_i = (p.x, p.y, t_i)$.*

**Proof.** We prove this observation by contradiction. Suppose that $p.w \neq p_i.w$. Then, $C(p) \neq C(p_i)$ (see Definition 4), which means that either there exists a cuboid which contains $p$ but does not contain $p_i$ or there exists a cuboid which contains $p_i$ but does not contain $p$. Both cases imply that an event rectangle must exist in $t_i < t < t_{i+1}$, which leads to the following contradiction: there is no event rectangle in $t_i < t < t_{i+1}$. $\square$

By Observation 1, the point-weight of $p$ is the same as the point-weight of $p_i$, the projection of $p$ onto the plane $t = t_i$. Thus, it is easy to compute a weighted cuboid partition if we know the point weight of every point on the plane $t = t_i$. To investigate the point-weight of each point on the plane $t = t_i$, we need the set of cuboids intersecting with the plane. This set, in fact, is the set of active cuboids at $t = t_i$. By taking the projection of each active cuboid onto the plane $t = t_i$, we can obtain a set $R$ of weighted rectangles. Then, we compute the *weighted rectangular partition* for $R$, a 2D version of the weighted cuboid partition. We give formal definitions of the weighted rectangle and the weighted rectangular partition as follows.

**Definition 7.** *(Weighted Rectangle) a weighted rectangle r is an axis-parallel rectangle with a real-valued weight r.o.*

**Definition 8.** *(Weighted Rectangular Partition) For the set R of weighted rectangles, a weighted rectangular partition RP is a set of weighted rectangles satisfying following conditions: (1) any pair of rectangles in RP are not overlapping, (2) the geometric union of rectangles in RP is the same as the geometric union of rectangles in R. (3) for each rectangle $r \in RP$, the point-weight of $p \in R$ is r.w.*

Now, we show how to find the local solution in the subspace $t_i \leq t < t_{i+1}$ by solving a 2D geometric problem. Suppose that we have an algorithm for computing a weighted rectangular partition $RP_i$ at $t = t_i$. $RP_i$ contains information of the point-weight of every point on $t = t_i$. Thus, we can obtain a subset of the weighted cuboid partition as follows: for each $r \in RP_i$, we create the weighted cuboid which is bounded by two plane $t = t_i$ and $t = t_{i+1}$, whose bottom rectangle is $r$, and whose weight is $r.w$. By selecting the set of cuboids with the maximum weight among the created cuboids, we can obtain the local solution in the subspace $t_i \leq t < t_{i+1}$.

By iterating this procedure until there are no event rectangles, the NPS algorithm can find the optimal solution. The outline of the NPS algorithm is summarized as follows. The NPS algorithm has a nested plane-sweep structure: the outer plane-sweep structure (lines 4–15 of Algorithm 1) and the inner plane-sweep algorithm (Algorithm 2). In the outer plane-sweep structure, the NPS algorithm calls the *inner plane-sweep algorithm* (line 14 of Algorithm 1) for each event rectangle encountered by the sweep-plane. The inner plane-sweep algorithm computes the weighted rectangular partition by using the plane-sweep approach on a two-dimensional plane with a data structure defined in Definition 9. By using the outcome of the inner plane-sweep algorithm, the NPS algorithm computes the local solution (lines 5–7 of Algorithm 1). When there are no event rectangles, the NPS can obtain the global optimal solution by selecting the set of cuboids with the maximum weight among the local solutions. To describe our algorithm in detail, we introduce a data structure in Section 4.2.2. Then, we explain the inner plane-sweep algorithm in Section 4.2.3 and the outer plane-sweep structure in Section 4.2.4.

---

**Algorithm 1:** Nested Plane-Sweep Algorithm

> **Data:** a set of weighted cuboids $C$
> **Result:** a set of cuboids with the maximum weight in $CP$, where $CP$ is a weighted cuboid
>            partition of $C$

1   $MaxSet \leftarrow \emptyset$ ;
2   $Y \leftarrow$ an empty array ;
3   $t_{prev} \leftarrow -\infty$ ;
4   **while** sweep-plane $p$ encounters an event rectangle $r$ **do**
5     |   $t \leftarrow p.t$ ;
6     |   $CP' \leftarrow$ GetLocalMaxWeightedCuboids($Y, t_{prev}, t$) ;
7     |   $MaxSet \leftarrow$ UpdateMaxSet($MaxSet \cup CP'$) ;
8     |   $t_{prev} \leftarrow t$ ;
9     |   **if** $r$ is the bottom rectangle of $c \in C$ **then**
10     |    |   $r.w \leftarrow c.w$ ;                                                    /* c becomes active */
11     |   **else if** $r$ is the top rectangle of $c \in C$ **then**
12     |    |   $r.w \leftarrow -1 \times c.w$ ;                                             /* c is no longer active */
13     |   **end**
14     |   $Y \leftarrow$ InnerPlaneSweep($Y_{prev}, r$) ;                                  /* Algorithm 2 */
15   **end**
16   $CP' \leftarrow$ GetLocalMaxWeightedCuboids($Y_{prev}, t_{prev}, t$) ;
17   $MaxSet \leftarrow$ UpdateMaxSet($MaxSet \cup CP'$) ;
18   **return** $MaxSet$ ;

---

---

**Algorithm 2:** The Inner Plane-Sweep Algorithm

---

**Data:** a sorted array of *ylines* $Y$, a weighted rectangle $r$

**Result:** the updated array $Y_{new}$ of *ylines*

1  $ly \leftarrow$ the lower bound of $r$ on the $y$-axis ;

2  $uy \leftarrow$ the upper bound of $r$ on the $y$-axis ;

3  **if** $Y$ *is empty* **then**

4  $\quad \mid \quad Y_{new} \leftarrow$ HandleCase0() ;

5  **else**

6  $\quad \mid \quad Y_{new} \leftarrow \varnothing$ ;

7  $\quad \mid \quad yLine_{lower} \leftarrow$ Y.GetGreatestLower($ly$) ;

8  $\quad \mid \quad$ **while** $yLine_{lower}.y \leq uy$ **do**

9  $\quad \mid \quad \quad \mid \quad yLine_{upper} \leftarrow$ Y.GetNext,Line() ;

10 $\quad \mid \quad \quad \mid \quad Y_{temp} \leftarrow$ HandleEachCase($yLine_{lower}, yLine_{upper}$) ;

11 $\quad \mid \quad \quad \mid \quad Y_{new} \leftarrow Y_{new} \cup Y_{temp}$ ;

12 $\quad \mid \quad \quad \mid \quad yLine \leftarrow yLine_{upper}$ ;

13 $\quad \mid \quad$ **end**

14 **end**

15 $Y_{new} \leftarrow Y \cup Y_{new}$ ;

16 $Y_{new} \leftarrow$ SortAndDeleteRedundant($Y_{new}$) ;

17 **return** $Y_{new}$ ;

---

### 4.2.2. Data Structure for Weighted Rectangles

We define a data structure named *y-line* for weighted rectangles as follows:

**Definition 9.** *(y-line) y-line* is a horizontal line parallel to the *x*-axis. Formally, *y-line* is defined as a tuple $< y, sp >$ where $y$ is the $y$-axis value and $sp$ is a sequence of *split-points*. Each *split-point* is a tuple $< x, sum >$, where $x$ is a partitioning point on the $x$-axis, and $sum$ is the sum of weights of rectangles intersecting with $(x, y)$.

We use a set of *y-lines* to represent a weighted rectangular partition on a plane as shown in the example of Figure 4. In Figure 4a, we have four *y-lines*. They define five weighted rectangles in Figure 4b. Specifically, each consecutive pair $(yline_i, yline_{i+1})$ of them defines a set of rectangles bounded by two horizontal line $y = yline_i.y$ and $y = yline_{i+1}.y$. For example, $yline_2$ and $yline_3$ defines a set of rectangles bounded by $y = y_2$ and $y = y_3$. Each consecutive pair $(sp_i, sp_{i+1})$ of split points in $sp$ of a *y-line* defines a rectangle which is bounded by two vertical line $x = sp_i.x$ and $x = sp_{i+1}.x$, and whose weight is $sp_i.sum$. For example, we show that the shaded rectangle in Figure 4b can be represented by two *y-lines* and their split points. Let us consider $yline_2 = < y_2, sp_2 >$, where $sp_2 = (sp_2^1, sp_2^2, sp_2^3, sp_2^4)$ as illustrated in Figure 4a. The shaded rectangle in Figure 4b is the rectangle whose bottom-left corner coordinate is $(sp_2^2.x, yline_2.y)$, whose top-right corner coordinate is $(sp_2^3.x, yline_3.y)$, and whose weight is $sp_2^2.sum$.

### 4.2.3. The Inner Plane-Sweep Algorithm

In Section 4.2.1, we described the inner plane-sweep algorithm as if it took a set $R$, the set of projections of all the active cuboids onto sweep-plane, as an input. It should be noted, however, that $R$ is a conceptual notation to help readers understand the overall algorithm rather than an actual input (we will explain the actual input later). In fact, we can obtain the weighted rectangular partition of $R$ by using existing methods [1,2]. However, using one of them in our NPS algorithm may cause redundant computations since only a small portion of the weighted rectangular partition of the current $R$ (at $t = t_{i+1}$) differ from that of the previous $R$ (at $t = t_i$). We can avoid those redundant

computations if we reuse the previously computed weighted rectangular partition and update it partially. (We implement an algorithm which uses the existing method proposed in [1] instead of our inner plane-sweep algorithm as a baseline algorithm and compare it with our NPS algorithm in Section 7.)

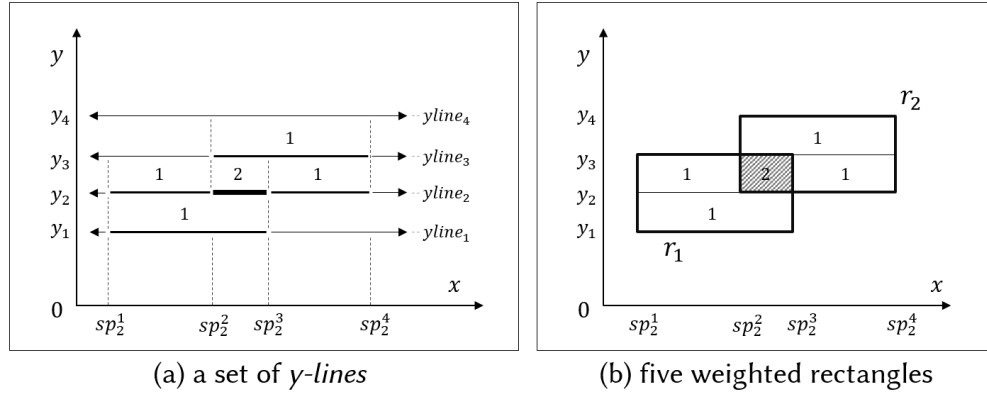

| (a) a set of *y-lines* | (b) five weighted rectangles |

**Figure 4.** An example of *y-lines*.

We show how the inner plane-sweep algorithm computes the weighted cuboid partition by reusing the previously computed result. Suppose that the sweep-plane encounters a new event rectangle $r$. Let $R_p$ be the previous $R$, $R_c$ be the current $R$ which is updated with $r$, and $Y$ be the set of *y-lines* representing the weighted rectangular partition of $R_p$. The inner-sweep algorithm takes $(Y, r)$ as input, and returns $Y_{new}$, which represents the weighted rectangular partition of $R_c$. While sweeping a line parallel to the *x*-axis from bottom to top, it incrementally computes $Y_{new}$. We call this line the sweep-line to avoid confusion with the 'sweep-plane' of the outer plane sweep structure. Since we do not have to consider *y-lines* which are not intersecting with $r$, the algorithm starts from the *y-line* right below (or on) the bottom line of $r$. Similarly, it terminates when the sweep-line encountered the *y-line* right above (or on) the top line of $r$.

We give a pseudo-code of the inner plane-sweep algorithm in Algorithm 2. Let us explain it with the running example. Figures 5–7 illustrate the step by step procedures of the NPS algorithm to solve the problem instance of Figure 2b. We draw a non-visited object as a transparent cuboid and a current visiting object as a half-transparent cuboid. Also, we illustrate thirteen cases that line 10 of Algorithm 2 can encounter during execution in Figure 8. Table 1 summarizes the update policy for each case.

Figure 5a depicts the scenario when the sweep-plane encounters $r_1$. Since $r_1$ is the first rectangle encountered by the sweep-plane, $Y$ is set to an empty array. This instance corresponds to case 0 in Figure 8. It is trivial to compute $Y_{new}$ for a single rectangle: the algorithm creates two *y-lines* (line 4 of Algorithm 2). Since there are no 'redundant *y-lines*', they just are sorted by their *y* values (line 16). We will explain redundant *y-lines* and the function 'SortAndDeleteRedundant' later. The output $Y_{new}$ is a sorted array of $\{yline_1, yline_2\}$, which represents this single rectangle as described in the bottom of Figure 5b.

Figure 6a depicts the scenario when the sweep-plane encounters $r_2$. As shown in the top of Figure 6b, $Y$ is given by $\{yline_1, yline_2\}$, the previously computed array of *y-lines*. The algorithm starts by searching the *y-line* right below $r_2$, namely $yline_1$ (line 7). Then, it fetches the next *y-line*, namely $yline_2$ (line 9). We now have a consecutive pair $(yline_1, yline_2)$ of *y-lines*. From now on, we denote the lower *y-line* by $yLine_{lower}$ and the upper *y-line* by $yLine_{upper}$ as described in Algorithm 2 (i.e., $yline_1=yLine_{lower}$, and $yline_2=yLine_{upper}$). As shown in Figure 8, this instance corresponds to the case 4 where $yLine_{lower}$ is below $r_2$ and $yLine_{upper}$ intersects with $r_2$. In this case, we need an additional *y-line* (namely, $yline_3$) to represent new area where point-weight changes (lines 10–11). To compute $yline_3$, the algorithm makes a copy of $yline_1$, and updates it by creating or updating *split-points* for $r_2$. The underlying principle of creating/updating *y-lines* is the same as that of [1,15]. In particular,

let $(sp_1, sp_2)$ be the first two element of $yline_3.sp$, where $sp_1 \; =< lx_1, 1 >$ and $sp_2 \; =< ux_1, 0 >$. Then, the algorithm updates it to be $(sp_3, sp_1, sp_4, sp_2)$ where $sp_3 \; =< lx_2, 1 >$, $sp_1 \; =< lx_1, 2 >$, $sp_4 \; =< ux_2, 1 >$, and $sp_2 \; =< ux_1, 0 >$ as shown in Figure 6b.

After creating $yline_3$, the sweep-line moves to the next *y-line*, namely $yline_2$. In other words, $yLine_{lower}$ becomes $yline_2$ (line 12). Since $yLine_{lower}.y$ is lower than $uy$ (line 8), the algorithm does not terminate, so it tries to fetch the next *y-line* (line 9). However, $yLine_{upper}$ is set to *null* because there are no *y-lines* any more. This instance corresponds to the case 3, so the algorithm updates $yline_2$ and creates a new *y-line* at $y = uy_2$ by the update policy of Table 1 as shown in the bottom of Figure 6 (we omit the detail update procedure due to the space limitation). Finally, the algorithm returns as an output the sorted array of four *y-lines*, which represents weighted rectangular partitions for $\{r_1, r_2\}$.

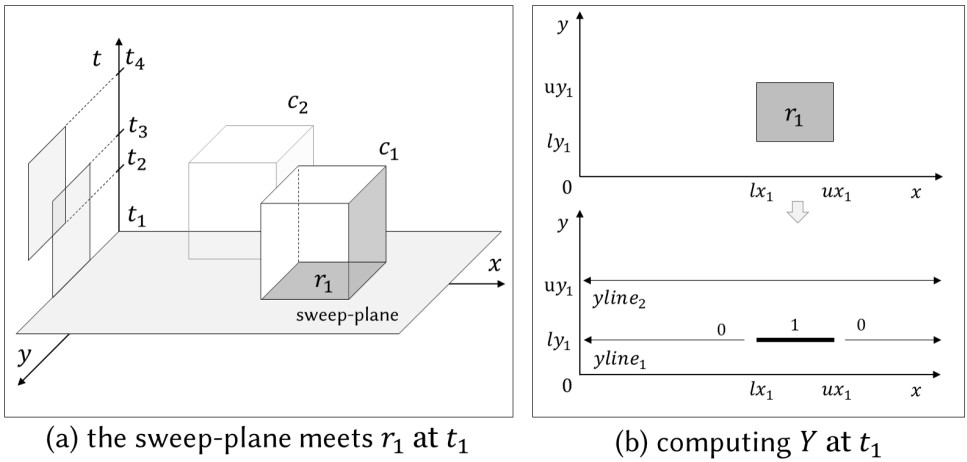

(a) the sweep-plane meets $r_1$ at $t_1$     (b) computing $Y$ at $t_1$

**Figure 5.** Sweep-plane encounters $r_1$ at $t_1$.

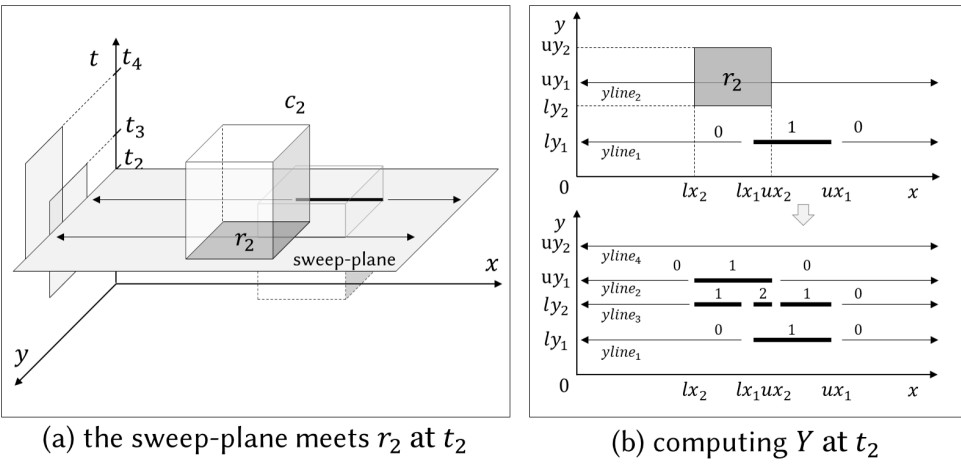

(a) the sweep-plane meets $r_2$ at $t_2$     (b) computing $Y$ at $t_2$

**Figure 6.** Sweep-plane encounters $r_2$ at $t_2$.

Figure 7a shows the scenario when the sweep-plane encounters $r_1'$, the top of $c_1$. The input is given by $r_1'$, and the array $Y$ of four *y-lines* representing the weighted rectangular partition of $\{r_1, r_2\}$. This example is different from the previous examples since it meets the top of a cuboid. In this case, we adopt the following trick instead of recomputing the weighted rectangular partition from the beginning. Setting the weight of $r_1'$ to $-1 \times r_1.w$, we invoke the inner plane sweep algorithm with input $(Y, r_1')$ to compute the weighted rectangular partition of $\{r_1, r_2, r_1'\}$. Since $r_1'$ cancels out the effect of $r_1$, we can obtain the weighted rectangular partition of $\{r_2\}$. The algorithm finally computes four *y-lines*, as shown in the bottom of Figure 7b. However, the dashed *y-lines* (i.e., $yline_1, yline_2$) are removed (line

16) because $yline_1$ has an empty sequence of *split-points*, and $yline_2$ has the same sequence of *split-points* as its previous *y-line* except for the *y*-value. If a *y-line* has the same configuration as its previous *y-line*, then it means that the point-weight of every point remains unchanged. If the bottommost *y-line* has an empty sequence, then it is meaningless. By a similar reason, if two consecutive *split-points* have the same sum of weight, then the upper *split-point* is removed.

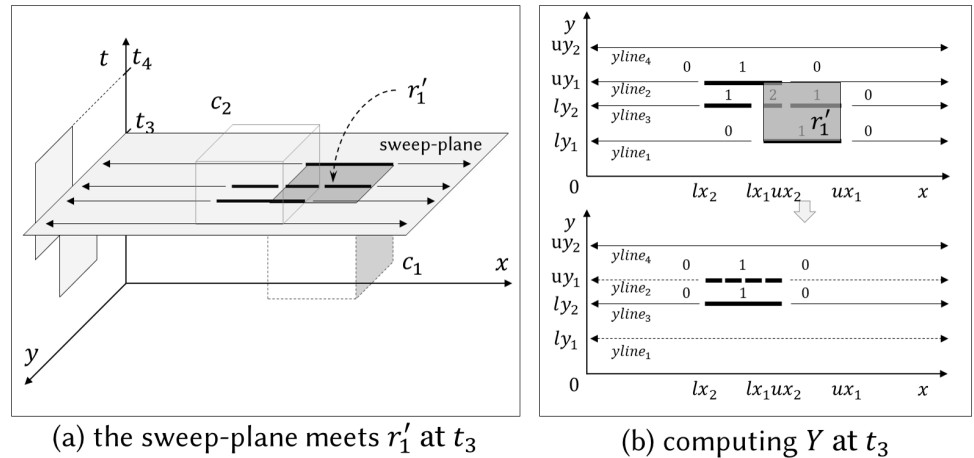

(a) the sweep-plane meets $r_1'$ at $t_3$　　　　　　　(b) computing $Y$ at $t_3$

**Figure 7.** Sweep-plane encounters $r_3$ at $t_3$.

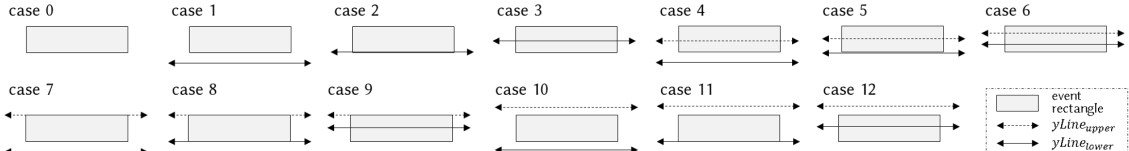

**Figure 8.** Cases of a pair of consecutive *y-lines*.

**Table 1.** Update policy for each case.

| Case # | *y-Line* Creation | Update $yLine_{lower}$ |
|---|---|---|
| case 0 | new *y-lines* for *ly*, *uy* | X |
| case 1 | new *y-lines* for *ly*, *uy* | X |
| case 2–3 | new *y-line* for *uy* | O |
| case 4 | new *y-line* for *ly* | X |
| case 5–6 | | O |
| case 7 | new *y-line* for *ly* | X |
| case 8–9 | | O |
| case 10 | new *y-lines* for *ly*, *uy* | X |
| case 11–12 | new *y-line* for *uy* | O |

### 4.2.4. The Outer Plane-Sweep Structure of NPS

The NPS algorithm scans each cuboid from bottom to top in its outer plane-sweep structure. While scanning each cuboid, it computes local solutions incrementally by invoking the inner plane-sweep algorithm (line 14 of Algorithm 1). First, the algorithm initializes variables *MaxSet*, *Y*, and $t_{prev}$: the set *MaxSet* of cuboids with the maximum weight is set to $\varnothing$, the array *Y* of *y-lines* is set to an empty array, and the previous timestamp $t_{prev}$ is set to $-\infty$ (lines 1–3 of Algorithm 1).

Suppose that the sweep-plane *p* encounters the i-th event rectangle *r* at $t = p.t$ (line 4) after it visited the (i-1)-th event rectangle $r_{prev}$ at $t_{prev}$. Since the sweep-plane moved upwards to encounter *r*, we need to update *MaxSet* before we update *Y*. To update *MaxSet*, the NPS algorithm computes $CP'$, a local solution in $t_{prev} \leq t < p.t$. Let $RP_{prev}$ be the weighted rectangular partition represented by $Y_{prev}$, previously computed at $t = t_{prev}$. Then, the function 'GetLocalMaxWeightedCuboids' finds

the local solution as follows (line 6): (1) for $r \in RP_{prev}$, it creates a cuboid whose bottom rectangle is $r$, which is bounded by two planes $t = t_{prev}$ and $t = p.t$, and whose weight is $r.w$, (2) and select the cuboid with the maximum weight in the created set of cuboids. The NPS algorithm updates *MaxSet* by selecting the cuboids with the maximum weight in $MaxSet \cup CP'$ (line 7). After updating *MaxSet*, it computes $Y$ which represents the weighted rectangular partition $RP_{current}$ for $R$, where $R$ is the set of bottom rectangles of active cuboids (lines 9–14).

The NPS algorithm iterates this process until there are no event rectangles. When the while-loop ends, the NPS algorithm updates *MaxSet* for the last event rectangle (lines 16–17), and returns it as a result. In the running example, the cuboid with weight 2, which is bounded by $x = lx_1$, $x = ux_2$, $y = ly_2$, $y = uy_1$, $t = t_2$, and $t = t_3$ is the final result (see Figures 6 and 7). Observe that this cuboid is identical to $c^*$ of Figure 2.

### 4.2.5. Analysis of the NPS Algorithm

The time complexity of the NPS algorithm is $O(|C| \log |C|) + O(|C| \log_2 \lambda_1 + \lambda_1 \lambda_2 |C|)$, where $\lambda_1$ is the maximum number of elements in $Y$ of *y-lines*, and $\lambda_2$ is the maximum number of elements in the sequence *yline.sp* of *split-points* among the created *y-lines* during the entire processing operation.

In particular, it requires $2|C| \times \log 2|C|$ operations for sorting cuboids for the outer plane-sweep structure. The inner plane-sweep algorithm has the complexity of $O(\log_2 \lambda_1 + \lambda_1 \lambda_2)$: it takes $O(\log_2 \lambda_1)$ to search $yLine_{lower}$ within a sorted array $Y$, and the upper bound of the number of iterations (lines 8–13 of Algorithm 2) is $\lambda_1 \lambda_2$, the upper bound of rectangles represented by *y-lines*. In addition, function 'GetLocalMaxWeightedCuboids' has the complexity of $O(\lambda_1 \lambda_2)$. Thus, the total time complexity for processing each event rectangle is $O(\log_2 \lambda_1 + \lambda_1 \lambda_2)$. Finally, the upper bound for the total time complexity of the NPS algorithm is $O(|C| \log |C|) + O(|C| \log_2 \lambda_1 + \lambda_1 \lambda_2 |C|)$.

The NPS algorithm is straightforward but has limited scalability. If it takes a large dataset or a large query cuboid, then it may create too many *y-lines*. Consequently, the time complexity dramatically increases since $\lambda_1$ and $\lambda_2$ increases.

To reduce computational overhead to search *y-lines*, we implement an improved NPS algorithm which manages *y-lines* with an in-memory height-balanced tree structure, named Rectangular Partition Tree (RP-Tree). Our RP-Tree satisfies the following characteristics: (1) it provides efficient retrieval of *y-lines* intersecting with a rectangle; (2) it provides an efficient in-order traversal from an arbitrary *y-line*. Since RP-Tree is similar to B+Tree [24,25], we omit the description of the improved NPS algorithm. Although we observed that the improved algorithm is more efficient than the original NPS algorithm, the improved one also has limited scalability because RP-Tree itself cannot reduce $\lambda_1$ or $\lambda_2$.

### 4.3. Divide and Conquer Algorithm

We can reduce the computational complexity by dividing the whole problem into independent smaller sub-problems. If each sub-problem deals with a small number of objects, it can be solved more effectively (this is because $\lambda_2$ is reduced). This approach is also known as the divide-and-conquer (DC) strategy, a natural way to cope with optimization problems in computational geometry. For example, Choi et al. [1,2] proposed a DC-based scalable algorithm for the two-dimensional MaxRS problem. Similar to [1,2], we propose an improved algorithm based on DC strategy.

### 4.3.1. Division Phase

First, our DC algorithm creates $c(o)$ for each $o \in O$. Then, it recursively divides the entire space into a set of subspaces until each subspace has a small enough number of cuboids. However, unlike in [1,2], the number of cuboids in each subspace does not have to fit in the main memory in our method necessarily. We will explain the reason for this later.

Specifically, to split the entire space $P$ into $M$ disjoint subspaces, it creates $M - 1$ planes perpendicular to the $x$-axis. Each cuboid $c$ in $C$ may be split into several cuboids by planes. As shown in Figure 9, if a subspace $S_i$ overlaps with a cuboid $c$, then $c$ is one of: (1) covered by $S_i$; (2) partially

intersecting with $S_i$; and (3) spanning $S_i$. It is trivial to handle the first and second cases: the intersection (of $c$ and $S_i$) is assigned to $S_i$. However, the third case is non-trivial because it may cause infinite recursion. Let us consider an extreme scenario, where every cuboid spans $S_i$. No matter how we divide $S_i$, there exists a subspace where the number of cuboids inside remains the same. A similar case, where a rectangle spans a subspace, is demonstrated in [1,2]. Authors of [1,2] solve this problem by preventing spanning objects from being passed to the next level recursion and processing them later in the merging phase together with intermediate results of sub-problems. Since spanning rectangles are excluded, it is possible to obtain a set of subspaces where each subspace has a small enough number of cuboids to fit into the main memory.

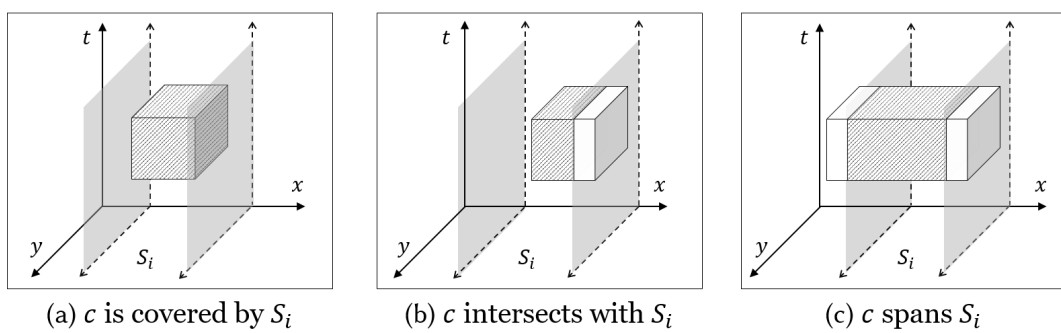

(a) $c$ is covered by $S_i$    (b) $c$ intersects with $S_i$    (c) $c$ spans $S_i$

**Figure 9.** Cases when subspace $S_i$ overlaps with $c$.

However, this method may not be the best solution in a spatiotemporal space for two critical reasons. First, the intermediate result of each sub-problem tends to be large to guarantee the correctness. Suppose that we have a sub-problem in a subspace $S_i$ where a small number of cuboids are inside. We can obtain $c_i^*$, the answer to the 3D MaxRS problem with input $C_i$, by using the NPS algorithm. Note that, if it is guaranteed that the $c_i^*$ is the solution in the subspace $S_i$, then we can return $c_i^*$ alone as an intermediate result. However, $c_i^*$ may not be the exact solution in $S_i$ because a spanning cuboid may exist in $S_i$. Thus, for every single pair of consecutive timestamp ($t_{prev}$, $t$), we have to report cuboids with the maximum weight in $CP'_{prev}$ (line 7 in Algorithm 1). In the merging phase, we have to consider every cuboid reported in each subspace and every spanning cuboid altogether to obtain the correct answer.

Second, we do not necessarily have to load all cuboids in a subspace into the main memory. We observed that memory accesses for input objects are not the bottleneck for computing the 3D MaxRS problem. In fact, each event rectangle is read once, and we can linearly scan all event rectangles after sorting them. Thus, accessing them from a sequential access memory (i.e., disk-storage) after sorting them does not significantly affect the overall execution time.

Due to these reasons, the number of cuboids in each subspace does not have to fit into the main memory necessarily. To prevent spanning rectangles from being created, our DC algorithm terminates recursion when the number of cuboids inside the subspace fits into the main memory, or the length of the space along the $x$-axis is equal to or smaller than $a$. Suppose that we obtain a set $S$ of subspaces. For each $S_i \in S$, it solves the 3D MaxRS problem with input $C_i$ of cuboids inside $S_i$. Although there may exist a subspace $S_i \in S$ such that all cuboids in $S_i$ cannot be loaded at the same time into main memory, we can compute the answer set efficiently by using the NPS algorithm.

### 4.3.2. Merging Phase

Our dividing method not only reduces the size of intermediate results significantly, but also makes the merging phase simple. Suppose that a subspace $S_i$ contains a set $C_i$ of weighted cuboids, and a set $A_i$ is the answer set of cuboids to the 3D MaxRS problem with input $C_i$. Unlike in [1,2], it is guaranteed that $A_i$ is the optimal solution in the subspace $S_i$ since there are no spanning objects. Thus,

the remaining task is selecting the set of cuboids with the maximum weight in $A$, where $A = \cup_{1 \le i \le M} A_i$, which is trivial to compute.

## 5. Algorithms for MaxStRS-AC

In this section, we show that the MaxStRS-AC problem is polynomially reducible to the 3D MaxRS problem. In addition, we introduce the baseline algorithm for the MaxStRS-AC problem and then present the extension of the baseline algorithm based on the divide-and-conquer approach.

### 5.1. Overview

The MaxStRS-AC problem tries to find the set of all points $p^* \in P$ such that:

$$p^* = \arg\max_{p \in P} \left( \sum_{o \in O(c(p))} o.w - \sum_{o \in O(c_{prev}(p))} o.w \right)$$

To examine a point $p$ whether it belongs to the answer set or not, we need to consider two different cuboids (i.e., $c(p)$ and $c(p_{prev})$) at the same time. Thus, it may seem that it is not feasible to solve the MaxStRS-AC problem with the plane-sweep approach at first glance. However, we observe an interesting property of the MaxStRS-AC problem: we can transform the MaxStRS-AC problem into the 3D MaxRS problem. We first present a novel idea to transform the MaxStRS-AC problem to the 3D MaxRS problem.

We illustrate the idea with a motivating example in Figure 10. Figure 10a shows a set $O$ of objects. Since $c(p)$ encloses $o_1$, $o_2$ and $c_{prev}(p)$ encloses $o_3$, $o_4$, $\sum_{o \in O(c(p))} o.w - \sum_{o \in O(c_{prev}(p))} o.w$ is equal to $o_1.w + o_2.w - o_3.w - o_4.w$. Observe that we need to consider both $c(p)$ and $c(p_{prev})$ to examine $p$ whether it belongs to the answer set or not. To transform the problem, we create a virtual object $o_i' = (o_i.x, o_i.y, o_i.t + \tau, -o_i.w)$ for each object $o_i$ as illustrated in Figure 10b. Let $O'$ be the set of created objects and $\hat{O}$ be the union of $O$ and $O'$ (i.e., $O \cup O'$). We now show that we no longer need to consider $c_{prev}(p)$ in the transformed problem. First, observe that $c(p)$ encloses not only $o_1$, $o_2$, but also $o_3'$, $o_4'$, as shown in Figure 10c. Thus, the sum of the weights of all the object in $c(p)$ is $o_1.w + o_2.w + o_3'.w + o_4'.w$, which is identical to $o_1.w + o_2.w - o_3.w - o_4.w$. In other words, $\sum_{o \in O(c(p))} o.w - \sum_{o \in O(c_{prev}(p))} o.w$ of Figure 10a is equivalent to $\sum_{o \in \hat{O}(c(p))} o.w$ of Figure 10c. Therefore, the goal of the MaxStRS-AC problem for $O$ is equivalent to the goal of the 3D MaxRS problem for $\hat{O}$.

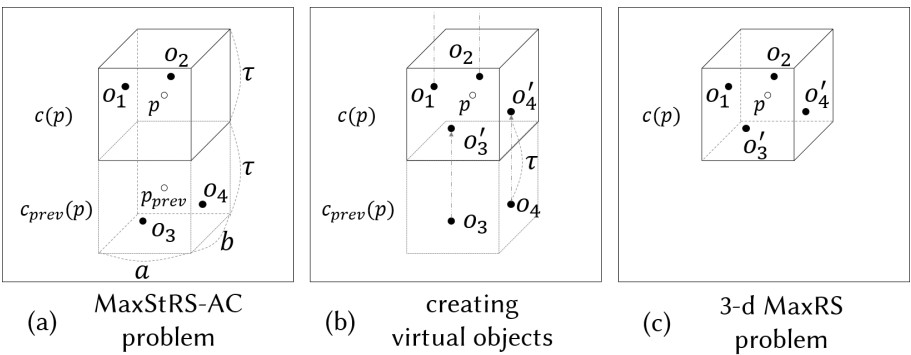

(a) MaxStRS-AC problem　　(b) creating virtual objects　　(c) 3-d MaxRS problem

**Figure 10.** A motivating example of transforming the MaxStRS-AC problem to the 3D-MaxRS problem.

Here, we give a mathematical justification for our method. To prove the reducibility, we need additional terminologies. For an object $o = (x, y, t, w)$, we use $\overline{o_{next}}$ to denote the virtual object $(o.x, o.y, o.t + \tau, -w)$. Similarly, for a set $O$ of objects, we use $\overline{O_{next}}$ to denote $\{\overline{o_{next}} | o \in O\}$. Lemma 1 and Theorem 1 show that we can transform the MaxStRS-AC problem into the 3D MaxRS problem.

**Lemma 1.** *Given O, the size $a \times b \times \tau$ of a query cuboid, the following equation holds for every p in the entire space:*

$$\sum_{o \in O(c_{prev}(p))} o.w = -1 \times \sum_{o \in \overline{O_{next}}(c(p))} o.w$$

**Proof.** We present the proof of *Lemma 1* in the Appendix A.　□

**Theorem 1.** *The answer to the MaxStRS-AC problem with input O and the size $a \times b \times \tau$ of a query cuboid is equivalent to the answer to the 3D-MaxRS problem with input $O \cup \overline{O_{next}}$ and the size $a \times b \times \tau$ of the query cuboid.*

**Proof.** The point $p^* \in P$ belongs to the answer set of the MaxStRS-AC problem if it satisfies the following equation:

$$p^* = \arg\max_{p \in P} \left( \sum_{o \in O(c(p))} o.w - \sum_{o \in O(c_{prev}(p))} o.w \right)$$

Using *Lemma 1*, we can safely rewrite $p^*$ as follows:

$$p^* = \arg\max_{p \in P} \left( \sum_{o \in O(c(p))} o.w + \sum_{o \in \overline{O_{next}}(c(p))} o.w \right) \tag{1}$$

$$= \arg\max_{p \in P} \left( \sum_{o \in \hat{O}(c(p))} o.w \right), \text{ where } \hat{O} = O \cup \overline{O_{next}} \tag{2}$$

Observe that Equation (2) is the form of the expression for the answer to the 3D MaxRS problem with input $O \cup \overline{O_{next}}$ and the size of $a \times b \times \tau$ of the query cuboid.　□

*5.2. Baseline Algorithm for MaxStRS-AC*

We present a two-phase nested plane-sweep based algorithm (NPS-AC) for the MaxStRS-AC problem. Suppose that we aim to solve the MaxStRS-AC problem with input $O$ and the size $a \times b \times \tau$ of a query cuboid. This algorithm consists of two phases: (1) generating a set $\hat{C}$ of cuboids such that $\hat{C} = C \cup \overline{C_{next}}$, where $C = \{c(o)|o \in O\}$, and $\overline{C_{next}} = \{c(o)|o \in \overline{O_{next}}\}$; (2) finding the answer set by using the NPS algorithm with input $\hat{C}$. Theorem 1 guarantees the correctness of this algorithm.

*5.3. Divide and Conquer Algorithm*

We introduce a divide-and-conquer based algorithm for the MaxStRS-AC (DC–AC) as an extension of the baseline algorithm. It also consists of two phases. In the first phase of each algorithm, $\hat{C} = C \cup \overline{C_{next}}$ is generated, as in that of the NPS-AC algorithm. In the second phase, the divide-and-conquer based algorithm for the MaxStRS-AC (DC–AC) finds the answer set based on the DC algorithm.

## 6. Algorithms for MaxStRS-RC

In this section, we show how to solve the MaxStRS-RC problem by computing a modified weighted cuboid partition, and then introduce the baseline algorithm for it. In addition, we present extensions of it, which is more scalable.

### 6.1. Overview

The MaxStRS-RC problem tries to find the set of all points $p^* \in P$ such that:

$$p^* = \underset{p \in P}{\arg\max} \left( \frac{\sum_{o \in O(c(p))} o.w - \sum_{o \in O(c_{prev}(p))} o.w}{\sum_{o \in O(c_{prev}(p))} o.w} \right) \tag{3}$$

To solve this problem, we introduce a new notation and an alternative version of the MaxStRS-RC problem as follows:

**Definition 10.** *(Modified Weighted Cuboid Partition) For the set C of weighted cuboids, a modified weighted cuboid partition MCP is a set of disjoint cuboids, where each cuboid c has two different weight $c.w_{change}$, $c.w_{prev}$, satisfying the following conditions:*

1. *MCP is a set of disjoint cuboids, that is, any pair in MCP are not overlapping.*
2. *The geometric union of cuboids in MCP is the same as the geometric union of cuboids in $C \cup \overline{C_{next}}$, where $\overline{C_{next}} = \{c(o)|o \in \overline{O_{next}}\}$.*
3. *For each cuboid $c \in MCP$, the point-weight of $p_{prev}$ for any $p \in c$ is the same as $c.w_{prev}$.*
4. *For any point p in each cuboid $c \in MCP$, the change between the point-weight of p and the past-point-weight of p (i.e., $p.w - p_{prev}.w$) is the same as $c.w_{change}$.*

**Definition 11.** *(Alternative version of the MaxStRS-RC). Let MCP be a modified weighted cuboid partition of the set C of weighted cuboids. Then, find $c^* \in MCP$ such that:*

$$c^* = \underset{c \in MCP}{\arg\max} \frac{c.w_{change}}{c.w_{prev}} \tag{4}$$

Similar to Section 4.1, the alternative version of the MaxStRS-RC problem is equivalent to the original MaxStRS-RC problem, and it is easy to solve the alternative version if we have a valid modified weighted partition of *C*.

Now, we show that computing a modified weighted cuboid partition can be transformed into computing a weighted cuboid partition. Suppose that $CP_{next}$ is a weighted cuboid partition of $\overline{C_{next}}$, and $CP_{change}$ is a weighted cuboid partition of $(C \cup \overline{C_{next}})$. For each point $p$ in a cuboid $c \in CP_{next}$, the point-weight of $p$ is same as $c.w = \sum_{o \in \overline{O_{next}}(c(p))} o.w$ (by Definitions 4 and 5). By Lemma 1. $\sum_{o \in \overline{O_{next}}(c(p))} o.w = -1 \times \sum_{o \in O(c_{prev}(p))} o.w$ holds. Thus, the point-weight of each point $p$ in a cuboid $c \in CP_{next}$ is $- \sum_{o \in O(c_{prev})} o.w$, which equals $-1 \times p_{prev}.w$ (i.e., $-1 \times$ denominator of Equation (4)). Similarly, the point-weight of each point $p$ equals $p.w - p_{prev}.w$ (i.e., numerator of Equation (4)). Therefore, if $CP_{next}$ and $CP_{change}$ are provided, we can obtain a modified weighted cuboid partition by partitioning them all together again. We describe the method in detail in the next subsection.

### 6.2. Baseline Algorithm for MaxStRS-RC

In this subsection, we propose a nested plane-sweep based algorithm (NPS-RC) for the MaxStRS-RC problem. The NPS-RC algorithm also consists of two phases. First, it creates set $\hat{C} = C \cup \overline{C_{next}}$. In the second phase, it computes a modified weighted cuboid partition $MCP$ and selects the set of cuboids with the maximum value of $\frac{c.w_{change}}{c.w_{prev}}$ in $MCP$.

The basic idea of the second phase of the NPS-RC algorithm is to compute $CP_{next}$ and $CP_{change}$ concurrently to obtain $MCP$ while scanning each cuboid in $C \cup \overline{C_{next}}$. To do so, we introduce a modified version of *y-line* as follows:

**Definition 12.** *(modified-y-line) modified-y-line, a horizontal line parallel to the x-axis, is defined as a tuple $< y, sp >$, where y is the y-axis value and sp is a sequence of modified-split-points. Each*

*modified-split-point* is a tuple $< x , sum_{change}, sum_{prev} >$ where $x$ is a partitioning point on the *x*-axis, $sum_{change}$ is the change between the weight-sums, and $sum_{prev}$ is the previous weight-sum.

Unlike an original *y-line*, a *modified-y-line* has a set of *modified-split-points* where each element contains two variables $sum_{change}$ and $sum_{prev}$ instead of a variables $sum$. Conceptually, $sum_{change}$ is used to represent $c.w_{change}$, and $sum_{prev}$ is used to represent $c.w_{prev}$. In particular, if the sweep-plane encounters an event rectangle $r$ which belongs to $C_{next}$, then the NPS-RC algorithm updates $sum_{change}$ and $sum_{prev}$ variables in *modified-y-lines* intersecting with $r$. On the other hand, if the sweep-plane encounters an event rectangle which belongs to $C$, then it updates $sum_{change}$ variables in *modified-y-lines* intersecting with $r$. Before updating the array of *modified-y-lines*, it computes a subset of *MCP*, and updates the current set of cuboids with the maximum value of $\frac{c.w_{change}}{c.w_{prev}}$. It iterates this procedure until there are no event rectangles.

### 6.3. Divide and Conquer Algorithm

We introduce a divide-and-conquer based algorithm (DC–RC) for the MaxStRS-RC problem. This algorithm first generates $\hat{C} = C \cup \overline{C_{next}}$. Then, it finds the answer set for the input $\hat{C}$ based on the divide-and-conquer strategy as follows. In the division phase, it divides the problem into sub-problems with the same manner in the division phase of the DC-Algorithm. Then, it solves each sub-problem using the NPS-RC algorithm. In the merging phase, it selects the set of cuboids maximizing $\frac{c.w_{change}}{c.w_{prev}}$.

## 7. Experimental Result

In this section, we evaluate the performance of the proposed algorithms and compare it with existing methods.

### 7.1. Experimental Setup

We evaluate the performance of algorithms both on synthetic and real datasets. First, we generate synthetic datasets in a spatiotemporal space of size $[0, 10000] \times [0, 10000] \times [0, 10000]$, where the first two dimensions correspond to space, and the third dimension corresponds to time. We use both Uniform distribution and Gaussian distribution for generating datasets in order to study how skewness affects the total response time. To create Gaussian-distributed datasets, we generate tuples by sampling a random number from a Gaussian distribution $N(\mu = 5000, \sigma = 1500)$ for each dimension. We create datasets of different sizes (10,000 to 500,000) for each data distribution. In addition, we collected a set of geo-tagged text data from Twitter using the Twitter Streaming API from 25 January 2018, to 28 June 2018. This real dataset contains 162,011 geo-tagged tweets that mentioned the keyword 'restaurant'.

We evaluate the performance of the NPS-based algorithm (optimized by RP-Tree) and DC-based algorithm comparing with existing methods by the response time and compare it against existing methods. Since no existing methods are directly applicable to the 3D MaxRS problem and the MaxStRSC problem, we extend existing algorithms for comparison as follows. For the 3D MaxRS problem, we extend the algorithm of [15], which tries to solve the max-enclosure problem in three-dimensional space in memory to solve the 3D MaxRS problem, and externalize it by employing STR-Tree [21] for better scalability. We call this algorithm 'UNIFIED' named after the title of their paper. Since the UNIFIED algorithm is somewhat outdated, we also implement another algorithm called 'BASELINE' Algorithm, as mentioned in Section 4.2.3. The BASELINE algorithm is similar to the NPS algorithm, but uses the algorithm proposed in [1] instead of our inner plane-sweep algorithm.

For the MaxStRS-AC problem, we extend the algorithm of [23]. We call this algorithm 'SURGE'. The original SURGE algorithm tries to continuously monitor the generalized version of the MaxStRS-AC problem over data streams. Thus, we first generate a data stream where each object in a set of historical spatiotemporal objects arrives in increasing order of timestamps and invoke the SURGE algorithm to handle this stream. By keeping track of the point with the maximum burst score, we can obtain at least one point which belongs to the answer set to the MaxStRS-AC problem. However,

note that it is impossible to extend the SURGE algorithm to solve the MaxStRS-RC problem since its pruning rules are no longer valid when we try to maximize $\frac{c.w_{change}}{c.w_{prev}}$.

We conduct experiments by varying parameters such as data cardinality, cuboid size, buffer size. We set the default experimental parameters as follows: data cardinality is $10^5$, a is 100, b is 100, t is 100, the buffer size is 2 MB, and the block size is 8 KB. We implemented all the algorithms in JAVA assuming that those algorithms will be deployed on environments based on a big data pipeline architecture such as Apache Hadoop and Spark, since programs written in JAVA are easily integrated into such architectures. We run all experiments on an Ubuntu (16.04 LTS) PC with Intel Xeon E5-1620 (3.6 Ghz Quad-core) and 32 GB memory.

### 7.2. Results of the 3D MaxRS Problem

We conduct various experiments for the 3D MaxRS problem on synthetic datasets as follows.

#### 7.2.1. Effect of Data Size

Experimental results for varying the size of a dataset are summarized in Figure 11, where the $y$-axis shows the response time in the log scale. Observe that our DC algorithm shows superior performance in all the cases. Although the NPS algorithm shows the worse performance than the DC algorithm, it is more efficient than the UNIFIED algorithm or the BASELINE algorithm. Observe that the response time of the DC algorithm does not rapidly increase when the size of the dataset increases, which means it is scalable. In addition, the DC algorithm is less sensitive to data skewness compared to the other algorithm.

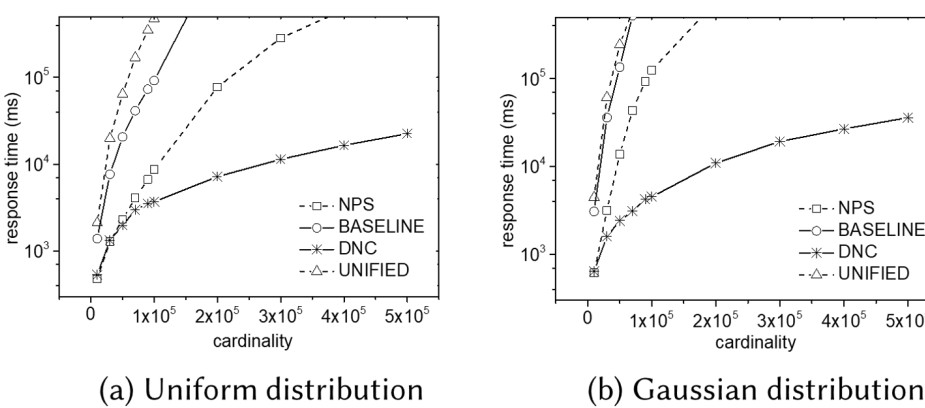

(a) Uniform distribution          (b) Gaussian distribution

**Figure 11.** Effect of data size in 3D MaxRS.

#### 7.2.2. Effect of Query Cuboid Size

We conduct two experiments to investigate the effect of query cuboid size. We do not plot the result of the BASELINE algorithm and the UNIFIED algorithm because they are much more inefficient than the NPS or the DC algorithm. Figure 12a summarizes the experimental results for varying the size of the spatial range (fixing the length of the temporal window to be 100). We use squares for the spatial range. In both distributions, the DC algorithm is more efficient than the NPS algorithm. While the performance of the NPS algorithm deteriorates as the size of the spatial range increases, the performance of the DC algorithm is stable. In addition, observe that the response time of the NPS algorithm severely increases in Gaussian distribution as the size of the spatial range increases. The reason for this is that the number of cuboids in the *weighted cuboid partition* increases extraordinarily in the Gaussian distribution. However, the performance of the DC algorithm in the Gaussian distribution is relatively stable since it reduces the computational cost by dividing the skewed space into several subspaces.

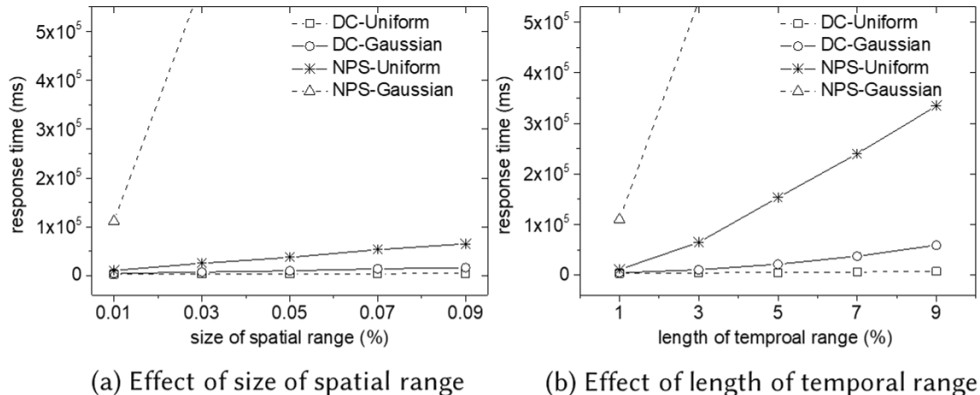

(a) Effect of size of spatial range

(b) Effect of length of temporal range

**Figure 12.** Effect of query cuboid size in 3D MaxRS.

Figure 12b summarizes the experimental results by varying the length of temporal range (i.e., $\tau$). The experimental results are similar to the results shown in Figure 12a. The performance of the DC algorithm is relatively stable, compared to that of the NPS algorithm.

### 7.2.3. Results Related to the Buffer Size

We conduct experiments to evaluate I/O-related efficiency. Since the other algorithms are in-memory algorithms, we only evaluate the performance of the DC algorithm by varying the size of (in-memory) buffer and the size of data (50k, 100k, and 150k) in both distributions. Although the UNIFIED algorithm takes advantage of a buffered data structure named STR-tree, we do not plot its performance because it is too inefficient.

As shown in Figure 13a, the response time of the DC algorithm decreases as we increase the buffer size in the uniform distribution. On the contrary, the response time of the DC algorithm tends to increase as we increase the buffer size in the Gaussian distribution. It means that using a larger size buffer does not always guarantee better performance, especially in the skewed data distribution. Recall that, if the number of cuboids in each subspace fits into the buffer, the DC algorithm stops to divide space in its division phase. If the buffer size increases, then the subspace of each sub-problem gets enlarged. Meanwhile, it may take more time to solve sub-problems in dense regions in the highly skewed data distribution. Thus, if the buffer size becomes too large in skewed data distribution, it may take much more time to solve some sub-problems in large subspaces.

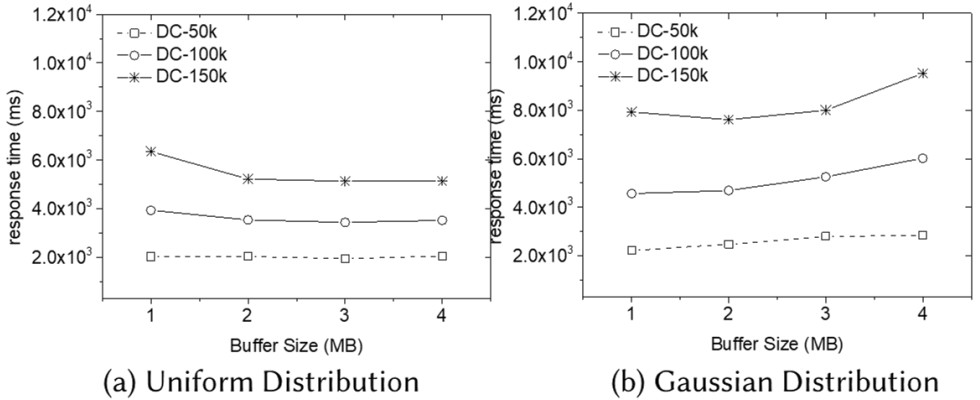

(a) Uniform Distribution

(b) Gaussian Distribution

**Figure 13.** Results related to the I/O in 3D MaxRS.

### 7.3. Results of the MaxStRSC Problem

We investigate the performance of the DC–AC algorithm for the MaxStRS-AC problem and the DC–RC algorithm for the MaxStRS-RC problem on synthetic data sets in both distributions by varying

the size of the dataset. As shown in Figure 14, we do not plot the performance of NPS-based algorithms since they are much more inefficient than DC-based algorithms. Instead, we plot the performance of the DC algorithm for the 3D MaxRS problem and the performance of the SURGE algorithm [23] for the MaxStRS-AC problem on the same input datasets for comparison.

The experimental results of the MaxStRSC problem are summarized in Figure 14, where the *y*-axis shows the response time in the log scale. The DC–AC algorithm has a slightly higher response time than the DC algorithm in both distributions because it has to deal with two times more objects (i.e., $|C \cup \overline{C_{next}}| = 2 \times |C|$). In addition, the DC–RC algorithm has a higher response time than the DC–AC algorithm in both distributions because the DC–RC algorithm computes a modified weighted cuboid partition, which is more complicated than the original weighted cuboid partition. The SURGE algorithm is about ten times slower than the DC–AC algorithm in both data distributions. We observe that the DC–RC algorithm is more sensitive to data skewness. We guess the reason for this as follows. Since the DC–RC algorithm needs additional geometric operations and memory space for managing *modified-y-lines*, the DC–RC algorithm takes more time than the DC–AC algorithm takes for the same dataset. This tendency becomes worse in the skewed data distribution.

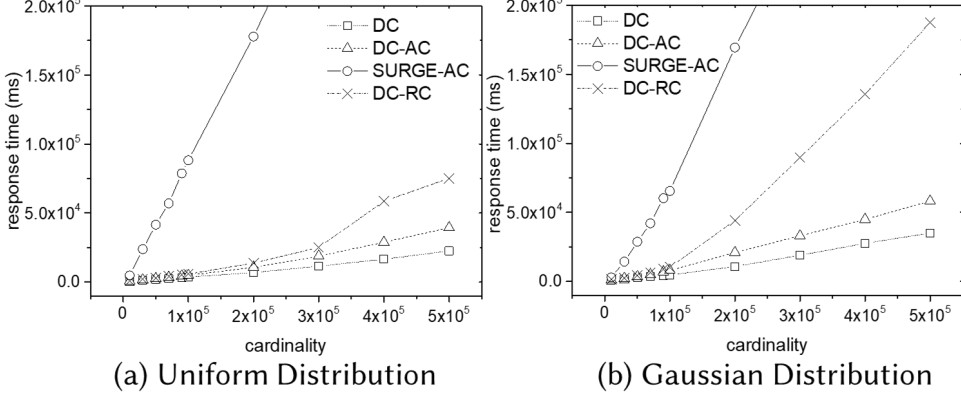

**Figure 14.** Effect of data size in MaxstRSC.

### 7.4. Experimental Results in Real Dataset

We conduct two experiments to investigate the effect of query cuboid size, as shown in Figure 15 in the real dataset collected from Twitter. We do not plot the result of the existing methods because they are much more inefficient than the NPS or the DC algorithms. For example, the response time of the SURGE algorithm with the size of query cuboid (0.1 km × 0.1 km × 100 s) on the Twitter dataset is more than one hour (5,263,782 ms).

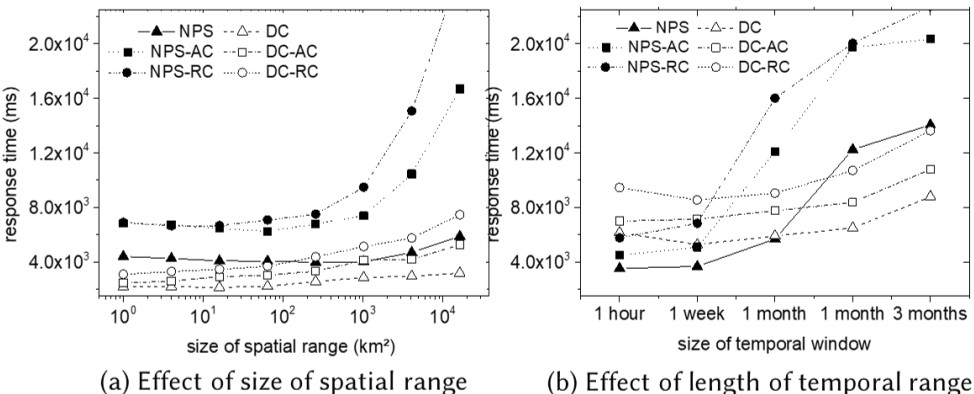

**Figure 15.** Experimental results in a real dataset.

Figure 15a summarizes the experimental results for varying the size of the spatial range (fixing the length of the temporal window to be one month). The DC-based algorithms are more efficient than NPS algorithms. While the performance of NPS-based algorithms deteriorates as the size of the spatial range increases, the performance of the DC-based algorithms is stable. Figure 15b shows the experimental results for varying the size of the temporal window (i.e., $\tau$), fixing the size of the spatial range to be 4 km $\times$ 4 km. For the short-term (less than one month), the NPS-based methods are usually better than the DC-based algorithms. However, the DC-based algorithms are more efficient than the NPS algorithms for a long-term period. The DC-based algorithm is less sensitive to the length of the temporal window.

## 8. Conclusions

In this paper, we introduce the 3D MaxRS problem and the MaxStRSC problem which can be used to find interesting spatiotemporal regions in a large historical spatiotemporal dataset. We first propose a nested plane sweep (NPS) algorithm for the 3D MaxRS problem, and then propose the divide-and-conquer algorithm (DC) for better scalability. In addition, we give a mathematical explanation for reducing the MaxStRSC problem to the 3D MaxRS problem and propose several algorithms for the MaxStRSC problem. The experimental results show that our DC-based algorithm is scalable and much more efficient than other algorithms (including existing methods) in general. As part of future work, we intend to build a system which executes analytical queries over the set of historical data and stream data in a distributed and parallel environment such as Apache Spark [26]. By executing the 3D MaxRS problem and the MaxStRSC problem in a distributed and parallel environment, we expect that it can provide more efficient and scalable support for spatiotemporal analysis over a large-scale spatiotemporal dataset.

**Author Contributions:** Conceptualization, W.C., K.-S.H., and S.-Y.J.; methodology, W.C.; validation, J.C. and K.P.; formal analysis, W.C.; investigation, W.C. and K.-S.H.; resources, J.C.; writing—original draft preparation, W.C.; writing—review and editing, S.-Y.J. and K.P.; visualization, W.C.; supervision, K.P.; project administration, S.-Y.J. and K.P.; funding acquisition, K.P.. All authors have read and agreed to the published version of the manuscript.

**Funding:** This work was supported by the National Research Foundation of Korea(NRF) grant funded by the Korea government(MSIP) (NRF-2018R1D1A1B07047618).

**Conflicts of Interest:** The authors declare no conflict of interest.

## Abbreviations

The following abbreviations are used in this manuscript:

| | |
|---|---|
| MaxRS | Maximum Range-Sum |
| MaxStRSC | Maximum Spatiotemporal Range-Sum Change |
| NPS | Nested Plane-Sweep algorithm |

## Appendix A. Proof of Lemma 1

We prove *Lemma 1* using the following notations. For an object $o = (x, y, t, w)$, we use $o_{next}$ to denote the object $(o.x, o.y, o.t + \tau, w)$, and $\overline{o_{next}}$ to denote the object $(o.x, o.y, o.t + \tau, -w)$. For a set of objects $O$, we denote $O_{next}$ to denote $\{o_{next} | o \in O\}$, and $\overline{O_{next}}$ to denote $\{\overline{o_{next}} | o \in O\}$.

**Proof.**

$$
\sum_{o \in O(c_{prev}(p))} o.w = \sum_{o \in \{o \in O | c_{prev}(p) \text{ contains } o\}} o.w
$$

$$
= \sum_{o \in \{o \in O | c(p) \text{ contains } o_{next}\}} o.w
$$

$$
= \sum_{o \in \{o \in O | c(p) \text{ contains } \overline{o_{next}}\}} (-1 \times o.w)
$$

$$
= \sum_{o \in \{o \in \overline{O_{next}} | c(p) \text{ contains } o\}} (-1 \times o.w)
$$

$$
= \sum_{o \in \overline{O_{next}}(c(p))} (-1 \times o.w)
$$

$$
= -1 \times \sum_{o \in \overline{O_{next}}(c(p))} o.w
$$

□

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
