# Peer review of "Scalable Algorithms for Maximizing Spatiotemporal Range Sum and Range Sum Change in Spatiotemporal Datasets"

_electronics, doi:10.3390/electronics9030514_

Round 1

Reviewer 1 Report

This manuscript investigates the spatiotemporal data analysis algorithms under varying parameters. The main contributions of the authors are new scalable solutions of MaxStRS problem modifications.

The idea and the advantages of the authors' approach are clearly described.

The manuscript is well-structured.

The experimental results verify that two maximum range sum change problems could be solved via the proposed algorithms.

Therefore, this work is meaningful and deserves to be published.

In Section “4.2.5. Analysis of the NPS algorithm”, please, add more details about time complexity.

In line 548, the measurement units of spatiotemporal space size are missing.

There are some spelling errors as follows:

  1. 7-8: “al gorithm” should become “al-gorithm”
  2. 11: “Also, None” -> “Also, none”
  3. 85: “Spatio-Temporal” -> “Spatiotemporal”
  4. 267: “(lines 14 of” -> “(line 14 of”
  5. 659: “Apache spark” -> “Apache Spark”

Other mistakes

  1. 23: The abbreviation “LBSNSs” is not defined.
  2. 228: The event rectangle r3 is missing in Figure 3 (b).
  3. 615: “Thus, if the buffer size too large in ..” – Please, edit this phrase.
  4. 633: “the DC-AC algorithm takes longer time than the DC-AC algorithm” -> “the DC-RC algorithm takes longer time than the DC-AC algorithm”.

Author Response

We appreciate Reviewer 1 for the comments and suggestions on our paper.

Point 1. In Section “4.2.5. Analysis of the NPS algorithm”, please, add more details about time complexity.

Response 1. We have added the time complexity of the NPS algorithm in detail.

Point 2. In line 548, the measurement units of spatiotemporal space size are missing.

Response 2. We agree that the previous description about the spatiotemporal space is confusing. However, we have regret to describe the measurement units of spatiotemporal space since the datasets we used were synthesized data sets where we can adjust parameters to fit real world data sets, regardless of the measurement units. 

As we mentioned, we acknowledge that the previous description should be improved for clarification. Especially, it was insufficient and unclear for users to tell which dimension corresponds space or which dimension corresponds time. Instead of mentioning unis, we have added the description about which dimension corresponds space or time.

Point 3. There are some spelling errors as follows:

  1. 7-8: “al gorithm” should become “al-gorithm”
  2. 11: “Also, None” -> “Also, none”
  3. 85: “Spatio-Temporal” -> “Spatiotemporal”
  4. 267: “(lines 14 of” -> “(line 14 of”
  5. 659: “Apache spark” -> “Apache Spark”

Response 3. We have corrected the errors raised by reviewer 1.

Please see the attachment for more details.

Reviewer 2 Report

This paper describes different algorithms for maximizing spatiotemporal range sum and range sum change problems.

The algorithms are detailed from the mathematical point of view and validated using both synthetic and real datasets.

The paper is not always easy to follow doe to the high number of definitions that sometimes seem unlined to the main text. I suggest to author to try to explain the same concepts adopting a different structure.

Concerning the experimental results, on page 18, authors describe the hardware and software used in the proposed work. Some details are missing since Ubuntu is the name of the operating system, but it is also necessary to provide the version of the system. Moreover, concerning Java, this choice is a bit strange. Why did the author chose this language? In this reviewer’s opinion, a language such as C++ could provide a better optimization and better performance than Java. I think that authors should discuss this aspect.

Minor points and typos:

- Please, do not use the capital letter after the colon.

- References 27, 28 and 29 are not cited in the text.

- On line 11, please change “None” with “none”.

- On line 23, please provide the full acronym of “LBSNSs”.

- On line 242, please change “Observation” with “Remark”.

- On line 314, please change “Figure 6 to 8” with “Figures 6 to 8”.

Author Response

We appreciate Reviewer 2 for the comments and suggestions on our paper.

Point 1. The paper is not always easy to follow doe to the high number of definitions that sometimes seem unlined to the main text. I suggest to author to try to explain the same concepts adopting a different structure.

Response 1. We admit that the previous manuscript gave a number of definitions, and we made some simple things hard to understand. We have modified the manuscript as follows: 

  1. We give the simple definitions of MaxStRS-AC and MaxStRS-RC in a bullet format, instead of a numbered definition format.
  2. We have added more description into some definitions (i.e., Weighted Cuboid Partition, Modified Weighted Cuboid Partition). For example, to make reader understand concepts better, we have paraphrase some conditions using a keyword such as 'disjoint cuboids' to briefly express 'any pair of cuboids in ~ are not overlapping). Also, we have reformatted these definitions for better readability.

Point 2. Some details are missing since Ubuntu is the name of the operating system, but it is also necessary to provide the version of the system

Response 2. We have added the version of the system (16.04 LTS)

Point 3. Moreover, concerning Java, this choice is a bit strange. Why did the author chose this language? In this reviewer’s opinion, a language such as C++ could provide a better optimization and better performance than Java. I think that authors should discuss this aspect.

Response 3. We acknowledge the point raised by the reviewer 2. We also think that C++ will provide a better optimization and better performance than java. However, we have assumed that the environment where our algorithm will be deployed would be based on a big data pipeline architecture such as Apache Hadoop and Spark. Programs written in JAVA are easily integrated into such architectures, and even can be extended into their API such as Map-Reduce framework or RDD with less efforts, since every element can be executed natively in rvm environment. This reason has driven us to conclude that conducting experiments on JAVA environment can provide more practical results than those on c++ environment. We have added the reason in the revised manuscript.

Point 4.

- Please, do not use the capital letter after the colon.

- References 27, 28 and 29 are not cited in the text.

- On line 11, please change “None” with “none”.

- On line 23, please provide the full acronym of “LBSNSs”.

- On line 242, please change “Observation” with “Remark”.

- On line 314, please change “Figure 6 to 8” with “Figures 6 to 8”.

Response 4.We have corrected the errors raised by reviewer 2.

Please see the attachment for more details.
